# Machine Learning Prediction Models for Mortality in Intensive Care Unit Patients with Lactic Acidosis

**DOI:** 10.3390/jcm10215021

**Published:** 2021-10-28

**Authors:** Pattharawin Pattharanitima, Charat Thongprayoon, Wisit Kaewput, Fawad Qureshi, Fahad Qureshi, Tananchai Petnak, Narat Srivali, Guido Gembillo, Oisin A. O’Corragain, Supavit Chesdachai, Saraschandra Vallabhajosyula, Pramod K. Guru, Michael A. Mao, Vesna D. Garovic, John J. Dillon, Wisit Cheungpasitporn

**Affiliations:** 1Department of Internal Medicine, Faculty of Medicine, Thammasat University, Pathum Thani 12121, Thailand; 2Division of Nephrology and Hypertension, Department of Medicine, Mayo Clinic, Rochester, MN 55905, USA; Qureshi.Fawad@mayo.edu (F.Q.); garovic.Vesna@mayo.edu (V.D.G.); dillon.John@mayo.edu (J.J.D.); 3Department of Military and Community Medicine, Phramongkutklao College of Medicine, Bangkok 10400, Thailand; wisitnephro@gmail.com; 4School of Medicine, University of Missouri-Kansas City, Kansas City, MO 64110, USA; fmaqureshi@gmail.com; 5Division of Pulmonary and Pulmonary Critical Care Medicine, Faculty of Medicine, Ramathibodi Hospital, Mahidol University, Bangkok 10400, Thailand; petnak@yahoo.com; 6Division of Pulmonary Medicine, St. Agnes Hospital, Baltimore, MD 21229, USA; nsrivali@gmail.com; 7Unit of Nephrology and Dialysis, Department of Clinical and Experimental Medicine, University of Messina, 98125 Messina, Italy; guidogembillo@live.it; 8Department of Thoracic Medicine and Surgery, Temple University Hospital, Philadelphia, PA 19140, USA; 109426469@umail.ucc.ie; 9Division of Infectious Disease, Department of Medicine, Mayo Clinic, Rochester, MN 55905, USA; s.chesdachai@gmail.com; 10Section of Cardiovascular Medicine, Department of Medicine, Wake Forest University School of Medicine, Winston-Salem, NC 27101, USA; svallabh@wakehealth.edu; 11Critical Care Medicine, Mayo Clinic, Jacksonville, FL 32224, USA; guru.pramod@mayo.edu; 12Division of Nephrology and Hypertension, Department of Medicine, Mayo Clinic, Jacksonville, FL 32224, USA; mao.michael@mayo.edu

**Keywords:** lactic acid, lactic acidosis, lactate, mortality, intensive care unit, machine learning, artificial intelligence, critical care, critical care medicine, nephrology, precision medicine, personalized medicine, individualized medicine

## Abstract

Background: Lactic acidosis is the most common cause of anion gap metabolic acidosis in the intensive care unit (ICU), associated with poor outcomes including mortality. We sought to compare machine learning (ML) approaches versus logistic regression analysis for prediction of mortality in lactic acidosis patients admitted to the ICU. Methods: We used the Medical Information Mart for Intensive Care (MIMIC-III) database to identify ICU adult patients with lactic acidosis (serum lactate ≥4 mmol/L). The outcome of interest was hospital mortality. We developed prediction models using four ML approaches consisting of random forest (RF), decision tree (DT), extreme gradient boosting (XGBoost), artificial neural network (ANN), and statistical modeling with forward stepwise logistic regression using the testing dataset. We then assessed model performance using area under the receiver operating characteristic curve (AUROC), accuracy, precision, error rate, Matthews correlation coefficient (MCC), F1 score, and assessed model calibration using the Brier score, in the independent testing dataset. Results: Of 1919 lactic acidosis ICU patients, 1535 and 384 were included in the training and testing dataset, respectively. Hospital mortality was 30%. RF had the highest AUROC at 0.83, followed by logistic regression 0.81, XGBoost 0.81, ANN 0.79, and DT 0.71. In addition, RF also had the highest accuracy (0.79), MCC (0.45), F1 score (0.56), and lowest error rate (21.4%). The RF model was the most well-calibrated. The Brier score for RF, DT, XGBoost, ANN, and multivariable logistic regression was 0.15, 0.19, 0.18, 0.19, and 0.16, respectively. The RF model outperformed multivariable logistic regression model, SOFA score (AUROC 0.74), SAP II score (AUROC 0.77), and Charlson score (AUROC 0.69). Conclusion: The ML prediction model using RF algorithm provided the highest predictive performance for hospital mortality among ICU patient with lactic acidosis.

## 1. Introduction

Lactic acidosis is the most common cause of anion gap metabolic acidosis in the intensive care unit (ICU) [1,2], and it is associated with poor clinical outcomes [1,3,4,5,6]. While ICU patients with lactic acidosis typically have classic type A lactic acidosis due to tissue hypoperfusion, certain patients with liver disease, cancer, or predisposing drug use (such as metformin, salicylate) can also have coexisting type B lactic acidosis [7,8]. Lactic acidosis (serum lactate ≥4 mmol/L) is associated with high mortality in various ICU settings, including sepsis, trauma, or cardiac surgeries [3,4,9,10,11,12,13,14,15,16,17,18,19,20,21,22,23,24,25]. Hence, the incorporation of ICU admission lactate levels in illness severity scores has been proven to improve predictive performance for in-hospital mortality [26,27]. 

Recently, artificial intelligence (AI) and machine learning (ML) have been increasingly utilized for precision medicine [28,29], including prediction of clinical outcomes among critically ill patients [30,31,32,33,34]. Due to the ability of ML to cope with nonlinear, complex, and multidimensional data [31,35], recent studies have demonstrated that ML approaches using ICU data provided high predictive performances that outperformed traditional analysis [32,33]. While the use of lactate levels in mortality prediction among critically ill patients has been investigated [26,27], data on mortality risk prediction among the subgroup of ICU patients with lactic acidosis are limited. Given the heterogeneity of impacts of lactic acidosis on clinical outcomes in a variety of different patient characteristics and ICU settings (such as lactic acidosis in patients with trauma, cardiac surgery, and septic shock) [9,10,11,12,13,14,15,16,17,18,19,20], an ML-based mortality prediction model for ICU patients with lactic acidosis can provide a novel individualized approach to clinical decision making for critically ill patients.

In this study, we aimed to develop and then assess various ML-based prediction model performances in predicting mortality of ICU patients with lactic acidosis in comparison to the traditional statistical model.

## 2. Methods

### 2.1. Patient Population

The Mayo Clinic Institutional Review Board approved this observational study (IRB number-21-009222). We used the Medical Information Mart for Intensive Care III (MIMIC III) database to conduct this study. MIMIC-III provides deidentified comprehensive clinical data from ICU patients at Beth Israel Deaconess Medical Center in Boston, Massachusetts, United States between 2001 and 2012 [36]. The database is widely accessible to researchers internationally under a data use agreement. If patients had multiple ICU admissions, we analyzed only the first admission.

Inclusion criteria were (1) age ≥18 years and (2) presence of lactic acidosis at ICU admission, defined as the first serum lactate measured within 48 h of ICU admission of ≥4.0 mmol/L. The exclusion criteria were (1) no serum lactate measurements within 48 h of ICU admission or (2) being admitted to the ICU for ≤24 h. 

### 2.2. Data Collection

We abstracted data on patient characteristics, comorbidities, vital signs, organ support, and laboratory results for prediction model development. As our goal was to develop and assess a prediction model for mortality in lactic acidosis patients based on the available data at the time of ICU admission, we only used data that were present within 48 h of ICU admission for analysis. When multiple values existed, we selected the closest vital sign or laboratory value to lactic acidosis occurrence. We excluded laboratory results with more than 10% missing data. Otherwise, we imputed missing data through multiple imputation using Random Forest (RF).

### 2.3. Model Development

In order to utilize ML models to predict the risk of in-hospital mortality in ICU patients with lactic acidosis, we followed the TRIPOD to build these ML models (Online Appendix A) [37]. Spearman’s rank correlation was applied to assess the separate correlation of variables in the dataset and demonstrate no significant correlations (Appendix A). Numeric data were normalized to have a standard deviation of 1 and a mean of 0 [38]. The overall study cohort was randomized into a training (80%) and testing dataset (20%) as per the Pareto principle [39]. We used the training dataset to develop ML models. The testing cohort was blinded to all methods until the final evaluation. As a reference model, we used multivariable logistic regression analysis. We conducted forward stepwise variable selection using criteria of *p* < 0.20 for entry cut-off. 

ML models include decision tree (DT), RF, extreme gradient boosting (XGBoost), and deep learning. RF and XGBoost are both DT ensemble algorithms [40,41]. However, RF forests rely on bagging, which is a democratic process to “elect” the best decision among the subgroups of trees [40]. XGBoost is based on a gradient descent–boosting process, which is an ensemble of weak learners that is reinforced depending on the quality of the assessment [41]. We used deep learning based on a multi-layer feedforward artificial neural network (ANN) that is trained with stochastic gradient descent using back-propagation.

For DT analysis, the number of terminal nodes was determined considering the scree plot showing the relationship between the tree size and coefficient of variance. The decision tree was pruned based on cross-validated error results using the complexity parameter associated with the minimal error (Appendix A). For the RF model, the number of trees was 500, which yielded the lowest error rate (Appendix A), and mtry value was calculated by the square root of the number of variables [42]. For XGBoost and ANN, we created a hyperparameter tuning grid to identify the best combination of hyperparameters using cross-validation methods [43]. Detailed hyperparameters are provided in the Online Appendix A.

### 2.4. Model Evaluation and Calibration

Model performance was assessed with area under the receiver operating characteristic curve (AUROC), accuracy, precision, error rate (ERR), Matthews correlation coefficient (MCC), and F1 score in the testing dataset [44,45,46]. The formula for each measure is provided in the Online Appendix A. The Brier score was used to evaluate model calibration [47].

### 2.5. Explanations of the Features in the ML-Based Prediction Model That Drive Patient-Specific Predictions of Mortality

After we identified ML model with highest predictive performances, we applied the Shapley additive explanations (SHAP) values to explain which features initiate patient-specific estimates. In addition, we also applied the local interpretable model-agnostic explanations (LIME) approach to approximate a complex nonlinear model to a linear model near variables of interest. 

### 2.6. Statistical Analysis

All analyses were performed using R, version 4.0.3 (RStudio, Inc., Boston, MA, USA; http://www.rstudio.com/ (accessed on 15 January 2021)). We used the “rpart” package for DT, “randomForest” and “randomForestExplainer” for RF, “caret” package for XGBoost and grid search, “h2o” package for ANN and SHAP, “LIME” for LIME, and the “missForest” package for missing data imputation [48].

## 3. Results

A total of 1919 ICU patients with lactic acidosis were eligible for analysis. Of these, 1535 and 384 were included in the training and testing dataset, respectively. Table 1 shows the clinical characteristics of patients in the training and testing datasets. Clinical characteristics between the training and testing datasets were comparable. Hospital mortality was also similar between training and testing datasets (29.8% vs. 29.7%; *p* = 0.97).

The ERRs and AUROCs of all ML models and the multivariable logistic regression model for mortality prediction in the test data set are shown in Table 2 and Figure 1. RF showed the lowest ERR (21.4%) and the highest accuracy (0.79), precision (0.72), MCC score (0.45), F1 score (0.56), and AUROC (0.83, 95% confidence interval (CI) 0.79–0.87). While the decision tree demonstrated a simple algorithm to follow (Figure 2), it had the highest ERR (26.7%) and lowest accuracy (0.73), precision (0.59), MCC score (0.30), F1 score (0.44), and AUROC (0.71, 95%CI 0.66–0.77) among all ML models (Figure 1).

The results of multivariable logistic regression analysis with stepwise variable selection are shown in Table 3. The AUROC of the multivariable logistic prediction model with forward stepwise variable selection was 0.81 (95%CI 0.79–0.83). We also compared our predictive models with Sequential Organ Failure Assessment (SOFA) score, Simplified Acute Physiology Score (SAPS II) score (acute severity score) and Charlson score (comorbidity score). The RF model outperformed the multivariable logistic regression model, SOFA score (AUROC 0.74), SAP II score (AUROC 0.77), and Charlson score (AUROC 0.69) (Table 2).

The RF model was the most well-calibrated among all models (Appendix A). The Brier score for RF, DT, XGBoost, ANN, and multivariable logistic regression was 0.15, 0.19, 0.18, 0.19 and 0.16, respectively (Table 2). Variable importance analysis of RF, the best model, was performed. The top important variables of RF combined the mean decrease in Gini (how much each variable decreases the node impurity), decrease in accuracy, and *p* values of the clinical indices include BUN, anion gap, lactate level, INR, pO2, phosphate level, PTT, platelet count, pH, and baseline eGFR (Figure 3).

To identify the features that influenced the prediction model the most, we depicted the SHAP summary plot of RF model (Figure 4) and the top 20 features of the prediction model. This plot depicts how high and low features’ values were in relation to SHAP values in the testing dataset. According to the prediction model, the higher the SHAP value of a feature, the higher probability of mortality occurring. Additionally, we applied LIME into RF model to illustrate the impact of key features at the individual level (Figure 5).

## 4. Discussion

Significant efforts have been invested into the development of predictive risk models of mortality for ICU patients. Traditional statistical models such as logistic regression analysis have been previously utilized to construct such prognostication tools [49,50,51,52]. In recent years, ML predictive algorithms have emerged as a method to handle high-dimensional, unstructured, and complex structure data [28,29,30,31,32,33,34]. In this study, we compared ML models and a conventional multivariable logistic regression model to assess the best-performing model for in-hospital mortality among ICU patients with lactic acidosis. The findings from our study suggest that the RF algorithm demonstrated superior performance in prediction of mortality among critically ill patients with lactic acidosis compared to other predictive tools. 

Modern ICUs and advances in electronic health records (EHRs) generate vast amounts of complex and multidimensional data that provide valuable information on patient outcomes. This has led to considerable advances in precision medicine [53]. While elevated serum lactate levels have been shown to be associated with increased mortality [54,55,56] and its incorporation improves predictive performance in traditional logistic regression models among critically ill patients [26,27], mortality risk prediction among the subgroup of ICU patients with lactic acidosis [54,55,56] is limited, especially utilizing ML approaches. Furthermore, patients with lactic acidosis are heterogenous and the impact of lactic acidosis on ICU mortality varies based on the clinical ICU setting, such as trauma, cardiac surgery, and sepsis [54,55,56]. Given the heterogeneity of ICU patients with lactic acidosis and the lack of adequate tools for patient-level prognostication, clinicians may often resort to subjective gestalt judgment, which is prone to bias [57]. Thus, we investigated whether ML methods improved mortality prognostication for ICU lactic acidosis in this study to improve precision medicine. Our best model was reached using the RF algorithm, which was associated with the highest AUROC and lowest ERR compared to all other models. We acknowledged that AUROC has several flaws [58], and thus, we also investigated other evaluation metrics including accuracy, precision, MCC, and F1 score. These confirmed the robustness of our RF prediction model. Finally, the findings that the predicted probabilities are close to the expected probability distribution supports that our RF model for mortality predication among ICU patients with lactic acidosis is well-calibrated.

RF is a widely used ML approach that can effectively predict outcomes [40]. It does this by utilizing additive combinations of trees that are built using different subsets of data and variables [42,59]. This nonparametric and nonlinear machine learning RF method can resist noise, and thus, it is expected to build accurate prediction models using aggregated data [40,60]. As a type of robust nonparametric model, RF can simulate complex relationships and it does not depend on data distribution, such as in logistic regression [40]. In addition, RF works well on large datasets, particularly when there are many categorical independent variables and unbalanced data [42]. On the other hand, the logistic regression analysis approach uses a generalized linear equation and the stepwise variable selection method is based on the likelihood ratio test to describe the directed dependencies among a set of variables. These approaches require that a number of statistical assumptions must be met. Thus, logistic regression analysis possesses inherent bias and, consequently, low variance due to the rigid nature from the shape of the line. Our RF prediction model also outperformed acute severity scores (SOFA and SAPS II scores) and comorbidities score (Charlson score) for prediction of ICU patients with lactic acidosis hospital mortality. In addition, this study provided important information on variables in our RF model including BUN, anion gap, lactate level, INR, pO2, phosphate level, PTT, platelet count, pH, and baseline eGFR. These variables in the RF model for critically ill patients with lactic acidosis using a MIMICIII database will help each institution develop their individualized RF model to better prognosticate mortality risk. Additionally, we applied model-agnostic approaches including feature relevant explanation through SHAP and local explanations through the LIME approach [61], which demonstrated how our RF model can be used to explain how each feature contributes to mortality prediction among patients with lactic acidosis.

Although our study includes a large sample size of ICU patients with lactic acidosis and ICU admission data, there are several important limitations. First, our models utilized data obtained during the time of ICU admission in order to prognosticate mortality risk in the early ICU course. Thus, events that markedly altered the prognosis for an individual patient were not included. In addition, our study is retrospective and based on the MIMIC III database, a large single-center tertiary care hospital in the United States. Hence, the model might have been influenced by the specific clinical guidelines, practice, and treatment decisions for that institution. A future validation study with the updated MIMIC-IV database and external validation studies of ML prediction models are needed.

## 5. Conclusions

In conclusion, an ML prediction model using the RF algorithm (available online as a shiny app at https://wisitc.shinyapps.io/RandomForestLacticAcid/ created on 15 September 2021)) provided the highest predictive performance for hospital mortality among ICU patients with lactic acidosis. While future external validation studies are required, the findings of our study provide support towards the utilization of RF algorithms to improve risk stratification among critically patients with lactic acidosis.

## Figures and Tables

**Figure 1 jcm-10-05021-f001:**
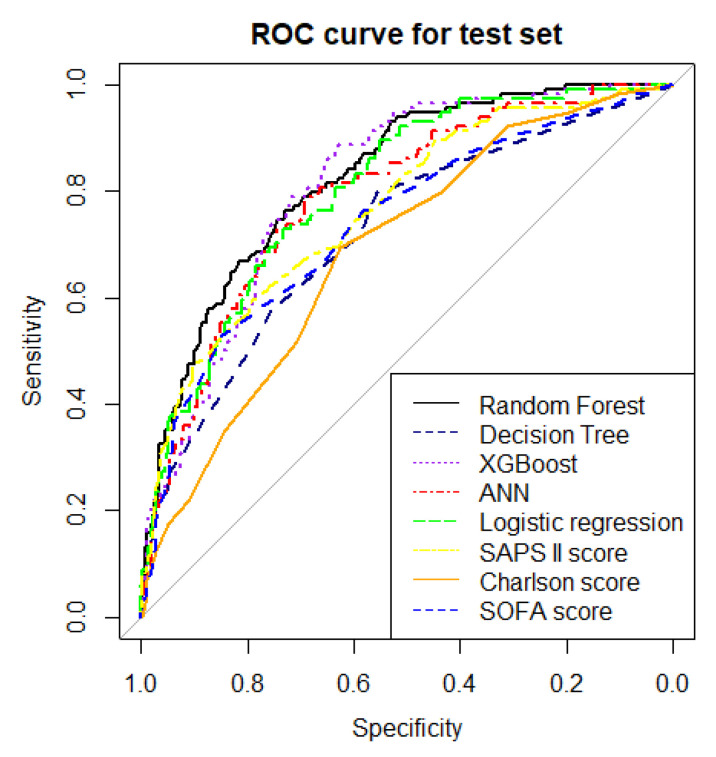
Comparison of AUROC among the different ML models, logistic regression model, SAPS II score, Charlson score, and SOFA score. ROC, operating characteristic.

**Figure 2 jcm-10-05021-f002:**
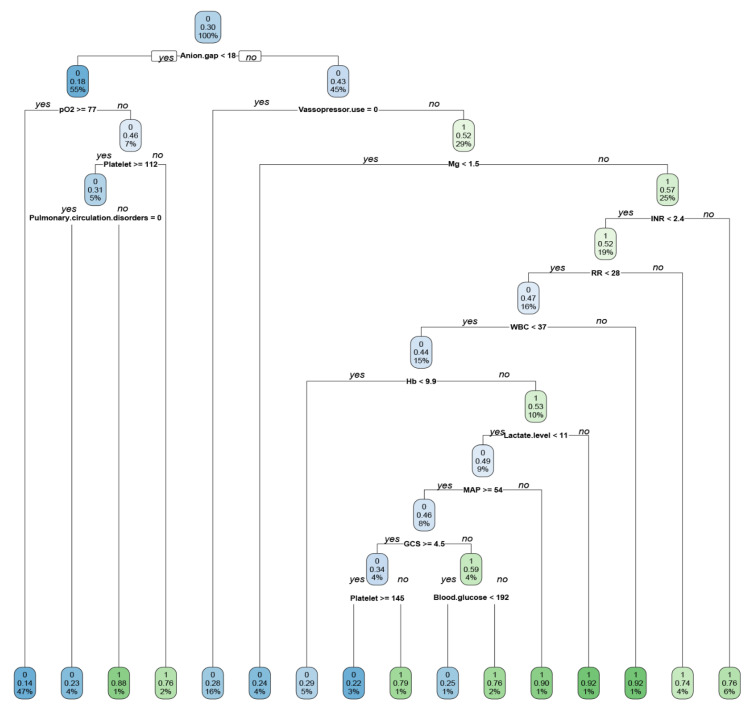
Simple decision tree model showing the classification of ICU patients with lactic acidosis who died (1) and did not (0) die during hospitalization. The numbers with two decimals in each cell mean the probability of mortality in each classification tree. The blue or green color becomes dense when it is more likely to die or not. The % number in the boxes denotes the percentage of patients with each discriminating variable from CART (Classification and Regression Tree) analysis. WBC, White Blood Cell; INR, International Normalized Ratio; RR, Respiratory Rate; MAP, Mean Arterial Pressure; GCS, Glasgow Coma Scale.

**Figure 3 jcm-10-05021-f003:**
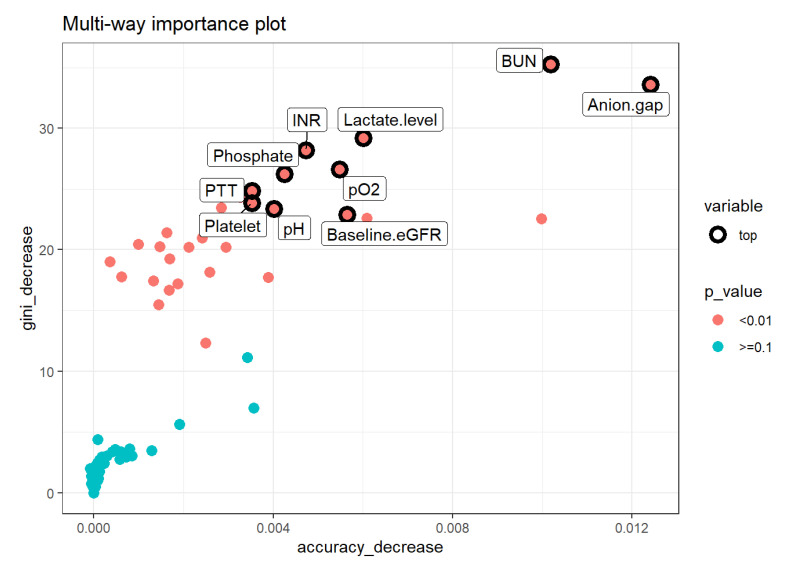
Multiway feature importance analysis of RF model combining the mean decrease in Gini, decrease in accuracy, and *p* values of the features (pink circle; *p* < 0.01). Abbreviations: BUN, blood urea nitrogen; eGFR, estimated glomerular filtration rate; pH, potential hydrogen; pO2, partial pressure of oxygen; PTT, partial thromboplastin time.

**Figure 4 jcm-10-05021-f004:**
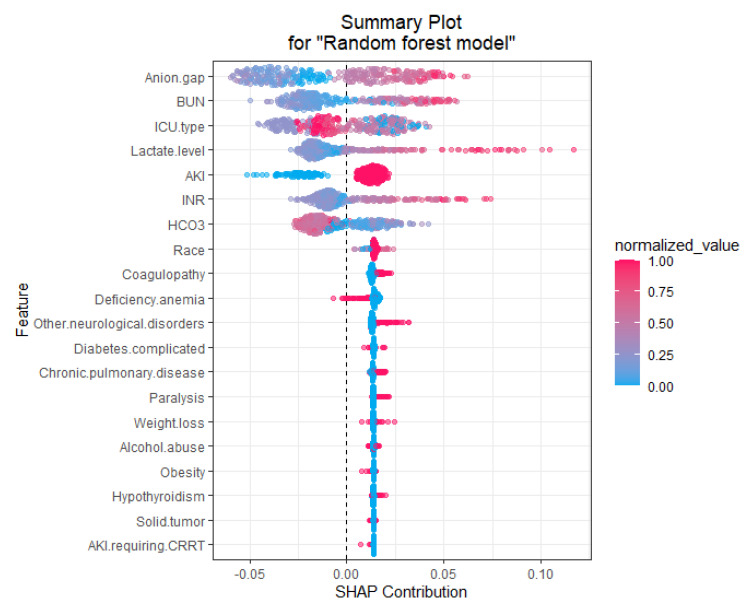
SHAP summary plot of the top 20 features of the RF model. The higher the SHAP value of a feature, the higher the probability of mortality. A dot is created for each feature attribution value for the model of individual patient. Each patient is allocated one dot on the line for each feature. Dots are colored according to the values of features for the respective patient and accumulate vertically to depict density. Red represents higher feature values, and blue represents lower feature values. Abbreviations: AKI, acute kidney injury; BUN, blood urea nitrogen; CRRT, continuous renal replacement therapy; HCO3, bicarbonate; ICU, intensive care unit; INR, international normalized ratio.

**Figure 5 jcm-10-05021-f005:**
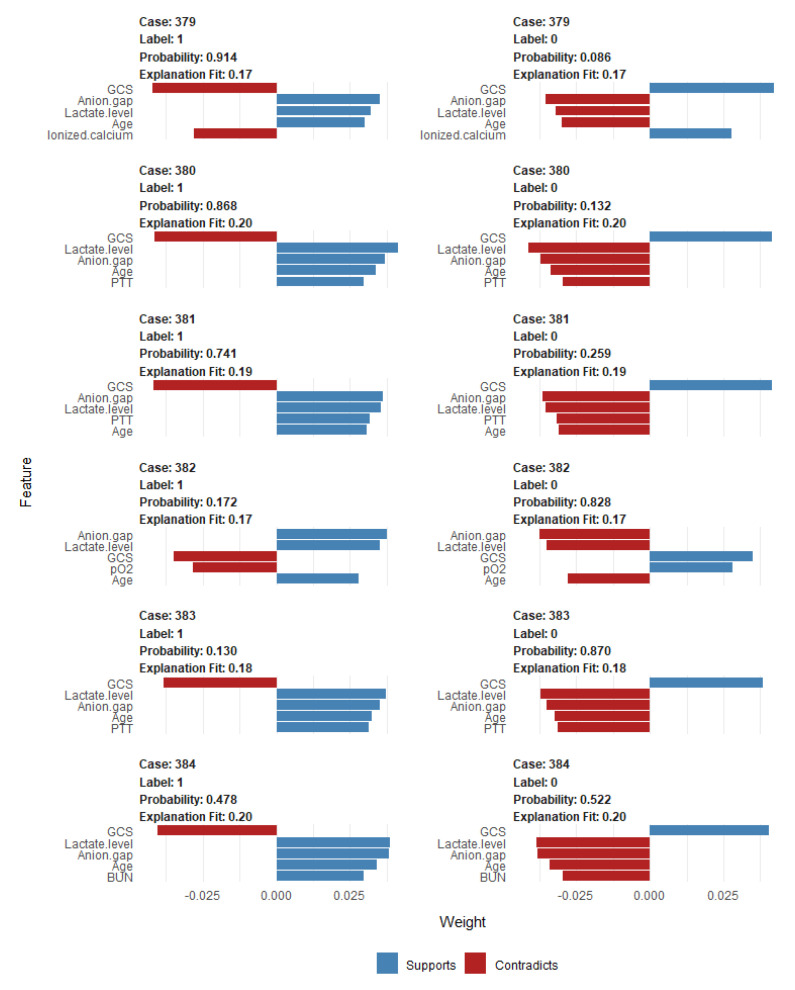
Local interpretable model explainer (LIME) for 6 individual cases (case# 379 to 384) from the testing dataset. Label “1” means prediction of mortality and label “0” means prediction of no mortality (survival). Probability shows the probability of the observation belong to the label “1” or “0”. The five most influential variables that best explain the linear model in that observation’s local region are provided and whether the variable causes an increase in the probability (supports/blue bar) or a decrease in the probability (contradicts/red bar). The x-axis shows how much each feature added or subtracted to the final probability value for the patient. Abbreviations: BUN, blood urea nitrogen; GCS, Glasgow Coma Scale; PTT, partial thromboplastin time; pO2, partial pressure of oxygen.

**Table 1 jcm-10-05021-t001:** Patient characteristics in the training and testing datasets.

Characteristics	All (*n* = 1919)	Training Set (*n* = 1535)	Testing Set (*n* = 384)	*p*-Value
Age (years)	61.8 ± 17.1	61.5 ± 17.0	63.0 ± 17.3	0.13
Male sex	1118 (58)	889 (58)	229 (60)	0.54
Race				0.85
White	1560 (81)	1246 (81)	314 (82)	
Black	152 (8)	124 (8)	28 (7)	
Hispanic	79 (4)	56 (4)	14 (4)	
Other	128 (7)	100 (7)	28 (7)	
ICU type				0.75
Cardiac ICU	206 (11)	164 (11)	42 (11)	
Cardiac surgery ICU	467 (24)	375 (24)	92 (24)	
Medical ICU	605 (32)	475 (31)	130 (34)	
Surgical ICU	295 (15)	243 (16)	52 (13)	
Trauma/surgical ICU	346 (18)	278 (18)	68 (18)	
Elixhauser Comorbidities				
Congestive heart failure	456 (24)	370 (24)	86 (22)	0.48
Valvular disease	352 (18)	282 (18)	70 (18)	0.95
Pulmonary circulation disorders	133 (7)	110 (7)	23 (6)	0.42
Peripheral vascular disease	286 (15)	227 (15)	59 (15)	0.78
Hypertension	884 (46)	694 (45)	190 (49)	0.13
Paralysis	55 (3)	36 (2)	19 (5)	0.006
Neurologic disorders	174 (9)	131 (9)	43 (11)	0.10
Chronic pulmonary disease	266 (14)	204 (13)	62 (16)	0.15
Uncomplicated diabetes	385 (20)	307 (20)	78 (20)	0.89
Complicated diabetes	73 (4)	61 (4)	12 (3)	0.44
Hypothyroidism	134 (7)	108 (7)	26 (7)	0.86
Liver disease	291 (15)	240 (16)	51 (13)	0.25
Peptic ulcer	1 (0.05)	1 (0.05)	0 (0)	0.62
AIDS/HIV	27 (1)	21 (1)	6 (2)	0.77
Lymphoma	52 (3)	41 (3)	11 (3)	0.83
Metastatic cancer	136 (7)	105 (7)	31 (8)	0.40
Solid tumor	128 (7)	103 (7)	25 (7)	0.97
Rheumatoid arthritis	41 (2)	36 (2)	5 (1)	0.20
Coagulopathy	500 (26)	395 (26)	105 (27)	0.52
Obesity	97 (5)	74 (5)	23 (6)	0.35
Weight loss	68 (4)	55 (4)	13 (3)	0.85
Fluid and electrolyte disorders	843 (44)	676 (44)	167 (43)	0.85
Blood loss anemia	36 (2)	29 (2)	7 (2)	0.93
Deficiency anemia	275 (14)	232 (15)	43 (11)	0.05
Alcohol abuse	199 (10)	168 (11)	31 (8)	0.10
Drug abuse	70 (4)	57 (4)	13 (3)	0.76
Psychosis	71 (4)	60 (4)	11 (3)	0.33
Depression	104 (5)	78 (5)	26 (7)	0.19
Chronic kidney disease	25 (1)	23 (1)	2(1)	0.13
Body weight (kg)	81.7 ± 21.0	81.9 ± 20.7	81.4 ± 22.1	0.72
Vital signs				
Temperature (F)	97.2 ± 2.2	97.2 ± 2.2	97.4 ± 2.0	0.23
Heart rate (per minutes)	97 ± 21	97 ± 21	97 ± 22	0.47
Systolic blood pressure (mmHg)	117 ± 26	117 ± 26	117 ± 24	0.86
Diastolic blood pressure (mmHg)	62 ± 15	62 ± 15	62 ± 15	0.91
Mean blood pressure (mmHg)	81 ± 21	82 ± 22	80 ± 18	0.32
Respiratory rate (per minutes)	17 ± 9	17 ± 9	17 ± 9	0.95
Oxygen saturation (%)	97 ± 5	97 ± 5	97 ± 5	0.18
Glasgow coma score	7.9 ± 4.9	8.3 ± 4.9	7.8 ± 4.9	0.05
Vasopressor use	1230 (64)	984 (64)	246 (64)	0.99
Ventilator use	1608 (84)	1285 (84)	323 (84)	0.85
Any renal replacement therapies	54 (3)	44 (3)	10 (3)	0.78
Hemodialysis	35 (2)	29 (2)	6 (2)	0.67
CRRT	22 (1)	18 (1)	4 (1)	0.83
Acute kidney injury	1401 (73)	1117 (73)	284 (74)	0.64
Laboratory data				
BUN (mg/dL)	27 ± 21	27 ± 20	28 ± 23	0.28
eGFR (mL/min/1.73 m^2^)	68 ± 31	68 ± 31	67 ± 29	0.65
Sodium (mEq/L)	138 ± 5	138 ± 6	139 ± 5	0.38
Potassium (mEq/L)	4.4 ± 0.9	4.3 ± 0.9	4.4 ± 0.9	0.45
Chloride (mEq/L)	106 ± 7	107 ± 6	106 ± 7	0.79
Bicarbonate (mEq/L)	20 ± 5	20 ± 5	20 ± 5	0.81
Anion gap (mEq/L)	18 ± 6	18 ± 5	18 ± 6	0.57
Total calcium (mg/dL)	8.2 ± 1.2	8.2 ± 1.2	8.2 ± 1.1	0.91
Ionized calcium (mmol/L)	1.1 ± 0.2	1.1 ± 0.2	1.1 ± 0.1	0.60
Phosphate (mg/dL)	4.1 ± 1.8	4.1 ± 1.7	4.2 ± 1.9	0.29
Magnesium (mg/dL)	1.9 ± 0.5	1.9 ± 0.5	2.0 ± 0.5	0.60
Lactate (mmol/L)	6.2 ± 2.6	6.2 ± 2.6	6.1 ± 2.5	0.45
Glucose (mg/dL)	179 ± 89	179 ± 88	180 ± 91	0.89
Hemoglobin (g/dL)	10.6 ± 2.3	10.6 ± 2.4	10.6 ± 2.3	0.98
WBC (109 cells/L)	14.1 ± 8.3	14.0 ± 8.6	14.2 ± 7.2	0.73
Platelet (109 cells/L)	170 ± 103	178 ± 102	187 ± 105	0.13
pH	7.31 ± 0.12	7.31 ± 0.12	7.31 ± 0.12	0.94
pCO2 (mmHg)	39 ± 11	39 ± 11	39 ± 11	0.96
pO2 (mmHg)	209 ± 133	209 ± 133	210 ± 134	0.80
INR	1.8 ± 1.0	1.8 ± 1.1	1.8 ± 1.0	0.98
PTT (second)	49 ± 30	49 ± 30	48 ± 31	0.82
Culture data				
Positive blood culture	197 (10)	158 (10)	39 (10)	0.94
Positive urine culture	205 (11)	171 (11)	34 (9)	0.19
Positive sputum culture	284 (15)	220 (14)	64 (17)	0.25
Hospital death	571 (30)	457 (30)	114 (30)	0.97

Abbreviations: AIDS, Acquired Immune Deficiency Syndrome; AKI, Acute Kidney Injury; BUN, Blood Urea Nitrogen; CCU, Coronary Care Unit; CHF, Chronic Heart Failure; Cl, Chloride; CRRT Continuous Renal Replacement Therapy; CSRU, Cardiac Surgery Recovery Unit; DBP, Diastolic Blood Pressure; eGFR, estimated Glomerular Filtration Rate; GCS, Glasgow Coma Scale; Hb Hemoglobin; HR, Heart Rate; ICU, Intensive Care Unit; IHD, Intermittent Hemodialysis; INR, International Normalized Ratio; K, Potassium; MAP, Mean Arterial Pressure; Mg, Magnesium; MICU, Medical Intensive Care Unit; Na, Sodium; pH, potential hydrogen; pCO2, partial pressure of carbon dioxide; pO2, partial pressure of oxygen; PT, Prothrombin time; PTT, Partial Thromboplastin Time; PVD, Peripheral vascular disease; RR, Respiratory Rate; RRT, Renal Replacement Therapy; SAPS II Score, Simplified Acute Physiology Score II; SPO2, Saturation of Peripheral Oxygen; Systolic Blood Pressure; SICU, Surgical Intensive Care Unit; WBC, White Blood Cell.

**Table 2 jcm-10-05021-t002:** Comparison of evaluation and calibration among the different models.

Model	Error Rate of Test Data Set	Accuracy	Precision	MCC	F1 Score	AUROC in the Test Set	Brier Score
Random forest model	21.4%	0.79	0.72	0.45	0.56	0.83 (0.79–0.87)	0.15
Decision tree	26.7%	0.73	0.59	0.30	0.44	0.71 (0.66–0.77)	0.19
XGBoost	25.0%	0.75	0.60	0.36	0.52	0.81 (0.76–0.85)	0.18
ANN	25.0%	0.75	0.67	0.33	0.42	0.79 (0.74–0.84)	0.19
Multivariable logistic regression	22.9%	0.77	0.67	0.41	0.54	0.81 (0.79–0.83)	0.16
SOFA score	25.5%	0.74	0.67	0.30	0.39	0.74 (0.68–0.80)	0.17
SAPS II score	23.2%	0.77	0.71	0.39	0.49	0.77 (0.71–0.82)	0.17
Charlson score	28.4%	0.72	0.73	0.16	0.13	0.69 (0.63–0.74)	0.19

MCC: worst value −1 and best value +1. F1 score, accuracy, and precision: worst value 0 and best value 1. The Brier score is a combined measure of discrimination and calibration that ranges between 0 and 1, where the best score is 0 and the worst is 1. ANN, artificial neural network; MCC, Matthews correlation coefficient; AUROC, area under the receiver operating characteristic curve; SOFA, Sequential Organ Failure Assessment; SAPS II, Simplified Acute Physiology Score.

**Table 3 jcm-10-05021-t003:** Development of multivariable logistic regression model to predict mortality using stepwise variable selection in the training dataset.

KERRYPNX	Univariate Analysis		Multivariable Analysis	
Characteristics	OR (95% CI)	*p*-Value	OR (95% CI)	*p*-Value
Age per 10 years	1.10 (1.03–1.17)	0.005	1.16 (1.07–1.26)	0.001
Male sex	0.93 (0.75–1.16)	0.52		
Race				
White	1 (reference)	1 (reference)		
Black	1.05 (0.70–1.56)	0.83		
Hispanic	0.37 (0.18–0.75)	0.006		
Other	0.93 (0.59–1.46)	0.75		
ICU type				
Cardiac ICU	1.21 (0.85–1.73)	0.30	0.73 (0.48–1.12)	0.15
Cardiac surgery ICU	0.21 (0.15–0.30)	<0.001	0.26 (0.16–0.42)	<0.001
Medical ICU	1 (reference)	1 (reference)	1 (reference)	1 (reference)
Surgical ICU	0.56 (0.40–0.77)	0.001	0.67 (0.45–1.00)	0.05
Trauma/surgical ICU	0.43 (0.31–0.60)	<0.001	0.82 (0.54–1.25)	0.35
Elixhauser Comorbidities	1.20 (0.93–1.54)	0.16		
Congestive heart failure	0.51 (0.37–0.71)	<0.001		
Valvular disease	1.21 (0.80–1.83)	0.36		
Pulmonary circulation disorders	0.80 (0.58–1.11)	0.18		
Peripheral vascular disease	0.75 (0.60–0.94)	0.01		
Hypertension	0.91 (0.43–1.89)	0.79		
Paralysis	1.17 (0.80–1.71)	0.43		
Neurologic disorders	0.98 (0.71–1.35)	0.90		
Chronic pulmonary disease	0.95 (0.73–1.26)	0.74		
Uncomplicated diabetes	0.69 (0.38–1.27)	0.24		
Complicated diabetes	0.81 (0.52–1.27)	0.37		
Hypothyroidism	1.80 (1.35–2.39)	<0.001		
Liver disease	2.17 (0.92–5.15)	0.08		
AIDS/HIV	3.12 (1.67–5.84)	<0.001		
Lymphoma	2.29 (1.53–3.41)	<0.001		
Metastatic cancer	0.74 (0.47–1.18)	0.21		
Solid tumor	0.91 (0.43–1.89)	0.79		
Rheumatoid arthritis	1.77 (1.39–2.25)	<0.001	1.96 (1.22–3.15)	0.005
Coagulopathy	0.49 (0.27–0.90)	0.02		
Obesity	0.97 (0.53–1.75)	0.91		
Weight loss	1.92 (1.54–2.40)	<0.001		
Fluid and electrolyte disorders	0.90 (0.39–2.04)	0.80	0.49 (0.25–0.99)	0.04
Blood loss anemia	0.61 (0.43–0.85)	0.003		
Deficiency anemia	0.94 (0.66–1.34)	0.72		
Alcohol abuse	1.00 (0.56–1.79)	0.99		
Drug abuse			0.50 (0.34–0.74)	<0.001
Psychosis	0.64 (0.34–1.20)	0.16		
Depression	0.92 (0.56–1.53)	0.76		
Chronic kidney disease	0.49 (0.17–1.46)	0.20		
Body weight per 5 kg	0.98 (0.96–1.01)	0.25		
Vital signs				
Temperature per 1 F	0.96 (0.91–1.01)	0.08		
Heart rate per 10 times/minute	1.11 (1.05–1.16)	<0.001		
Systolic per 10 mmHg	0.92 (0.88–0.96)	<0.001		
Diastolic BP per 5 mmHg	0.97 (0.93–1.01)	0.09	1.05 (1.01–1.10)	0.03
Mean BP per 5 mmHg	0.97 (0.94–0.99)	0.02		
Respiratory rate per 1 time/minute	1.05 (1.04–1.06)	<0.001	1.02 (1.00–1.03)	0.04
Oxygen saturation per 1 percent	0.93 (0.91–0.95)	<0.001		
Glasgow coma score per 1 unit	1.00 (0.98–1.02)	0.91		
Vasopressor use	2.19 (1.71–2.79)	<0.001	2.11 (1.54–2.89)	<0.001
Ventilator use	1.31 (0.96–1.79)	0.09	1.81 (1.21–2.70)	0.004
Any renal replacement therapies	1.36 (0.73–2.54)	0.33		
Hemodialysis	1.68 (0.80–3.55)	0.17		
CRRT	0.91 (0.32–2.56)	0.85		
Acute kidney injury	3.45 (2.55–4.67)	<0.001	2.10 (1.49–2.96)	<0.001
Laboratory data				
BUN per 1 mg/dL	1.03 (1.02–1.03)	<0.001		
eGFR per 10 mL/min/1.73 m^2^	0.90 (0.87–0.94)	<0.001		
Sodium per 1 mEq/L	1.00 (0.98–1.02)	0.72		
Potassium per 1 mEq/L	1.03 (0.92–1.16)	0.60		
Chloride per 1 mEq/L	0.95 (0.94–0.97)	<0.001	0.97 (0.95–0.99)	0.01
Bicarbonate per 1 mEq/L	0.90 (0.87–0.92)	<0.001		
Anion gap per 1 mEq/L	1.14 (1.12–1.17)	<0.001	1.04 (1.01–1.08)	0.009
Total calcium per 1 mg/dL	0.88 (0.80–0.96)	0.006		
Ionized calcium per 1 mmol/L	0.06 (0.03–0.13)	0.06	0.19 (0.08–0.46)	<0.001
Phosphate per 1 mg/dL	1.29 (1.21–1.37)	<0.001		
Magnesium per 1 mg/dL	1.48 (1.21–1.81)	<0.001	1.54 (1.18–2.02)	0.002
Lactate per 1 mmol/L	1.25 (1.20–1.31)	<0.001	1.11 (1.04–1.17)	0.001
Glucose per 1 mg/dL	1.00 (1.00–1.00)	0.14		
Hemoglobin per 1 g/dL	1.06 (1.01–1.11)	0.02		
WBC per 109 cells/L	1.01 (1.00–1.02)	0.13		
Platelet per 109 cells/L	1.00 (1.00–1.00)	0.12		
pH per 1 unit	0.04 (0.02–0.10)	<0.001		
pCO2 per 1 mmHg	0.99 (0.98–0.99)	0.04		
pO2 per 1 mmHg	1.00 (1.00–1.00)	<0.001	0.99 (0.99–1.00)	0.004
INR per 1 unit	1.62 (1.43–1.84)	<0.001	1.17 (1.03–1.33)	0.02
PTT per 1 s	1.01 (1.01–1.01)	<0.001	1.01 (1.00–1.01)	0.003
Culture data				
Positive blood culture	2.49 (1.79–3.48)	<0.001		
Positive urine culture	2.05 (1.53–2.74)	<0.001		
Positive sputum culture	1.90 (1.37–2.63)	<0.001		

Abbreviations: AIDS, Acquired Immune Deficiency Syndrome; AKI, Acute Kidney Injury; BUN, Blood Urea Nitrogen; CCU, Coronary Care Unit; CHF, Chronic Heart Failure; Cl, Chloride; CRRT Continuous Renal Replacement Therapy; CSRU, Cardiac Surgery Recovery Unit; DBP, Diastolic Blood Pressure; eGFR, estimated Glomerular Filtration Rate; GCS, Glasgow Coma Scale; Hb Hemoglobin; HR, Heart Rate; ICU, Intensive Care Unit; IHD, Intermittent Hemodialysis; INR, International Normalized Ratio; K, Potassium; MAP, Mean Arterial Pressure; Mg, Magnesium; MICU, Medical Intensive Care Unit; Na, Sodium; pH, potential hydrogen; pCO2, partial pressure of carbon dioxide; pO2, partial pressure of oxygen; PT, Prothrombin time; PTT, Partial Thromboplastin Time; PVD, Peripheral vascular disease; RR, Respiratory Rate; RRT, Renal Replacement Therapy; SPO2, Saturation of Peripheral Oxygen; Systolic Blood Pressure; SICU, Surgical Intensive Care Unit; WBC, White Blood Cell.

## Data Availability

Data are available upon reasonable request to the corresponding author.

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
