# Peer review of "Machine Learning Prediction Models for Mortality in Intensive Care Unit Patients with Lactic Acidosis"

_jcm, 2021, doi:10.3390/jcm10215021_

Round 1

Reviewer 1 Report

There are several comments:

  1. the MIMIC-III is outdated, which has patients decades ago and the clinical management of these patients may be different from current practice. Why not use the most updated MIMIC-IV?
  2. The models are not validated in external dataset, which can be prone to overfitting. the applicability of the models for excternal dataset is unknown.
  3. The rationale to focus on acidosis patients is not well described. why not for the whole ICU population? are these machine learning models more suitable for acidosis patients?
  4. "

    Given the heterogeneity of patients with lactic acidosis on ICU admission 

    "---I do not think this is specific for acidosis patients.
  5. The models should compare to several conventional severity scores; otherwise, the clinicians will not choose to use such complex ML models.
  6. The interpretability of the models should be further explored using LIME, SHARPLY and so on.

Author Response

Response to Reviewer#1

Comment #1

the MIMIC-III is outdated, which has patients decades ago and the clinical management of these patients may be different from current practice. Why not use the most updated MIMIC-IV? The models are not validated in external dataset, which can be prone to overfitting. the applicability of the models for external dataset is unknown.

Response: We thank you for reviewing our manuscript and for your critical evaluation. We appreciate the reviewer’s comment. The reviewer raised very important point and we are in the process of data preparation and preprocessing of MIMIC-IV dataset. We created various machine learning models and traditional logistic regression model in MIMIC-III dataset, and we then plan to validate our models in MIMIC-IV and our ICU dataset. We agree with this important point and thus added this important point in the discussion of our study. Additionally, we also deployed our RF prediction model to R shiny to increase visibility and also for other investigators to be able to utilize model for external validation. https://wisitc.shinyapps.io/RandomForestLacticAcid/

The following text has been added as suggested.

Future validation study in updated MIMIC-IV database and external validation studies of ML prediction models are needed.

“In conclusion, a ML prediction model using the RF algorithm (available online as a shiny app at https://wisitc.shinyapps.io/RandomForestLacticAcid/) provided the highest predictive performance for hospital mortality among ICU patient with lactic acidosis. While future external validation studies are required, the findings of our study provide support towards the utilization of RF algorithms to improve risk stratification among critically patients with lactic acidosis.”  

Comment #2

The rationale to focus on acidosis patients is not well described. why not for the whole ICU population? are these machine learning models more suitable for acidosis patients?

Response: We appreciate the reviewer’s important point. We additionally revised our background in the abstract, and also the introduction part to better explain the rationale of the study. While lactic acidosis is associated with poor outcomes in ICU, the associated mortality risks are different in various patient characteristics and ICU settings (such as trauma, cardiac surgery, and septic shock), and thus, a ML-based mortality prediction model for ICU patients with lactic acidosis can provide a novel individualized approach to clinical decision making for critically ill patients.
Lactic acidosis is the most common cause of anion gap metabolic acidosis in the intensive care unit (ICU), associated with poor outcomes including mortality. We sought to compare machine learning (ML) approaches versus logistic regression analysis for prediction of mortality in lactic acidosis patients admitted to the ICU.”

“Given the heterogeneity of patients with lactic acidosis on ICU admission and the associations impacts of lactic acidosis on clinical outcomes in a variety of different patient characteristics and ICU settings (such as lactic acidosis in patients with trauma, cardiac surgery, and septic shock) [9-20], a ML-based mortality prediction model for ICU patients with lactic acidosis can provide a novel individualized approach to clinical decision making for critically ill patients.”

Comment #3

" Given the heterogeneity of patients with lactic acidosis on ICU admission"---I do not think this is specific for acidosis patients.

Response: We appreciate the reviewer’s important comment. We agree and thus we have removed this sentence and have additionally revised as reviewer’s suggestion.

Given the heterogeneity of impacts of lactic acidosis on clinical outcomes in a variety of different patient characteristics and ICU settings (such as lactic acidosis in patients with trauma, cardiac surgery, and septic shock) [9-20], a ML-based mortality prediction model for ICU patients with lactic acidosis can provide a novel individualized approach to clinical decision making for critically ill patients.

Comment #4

The models should compare to several conventional severity scores; otherwise, the clinicians will not choose to use such complex ML models.

Response: We agree with the reviewer’s important comment. In addition to SAPS II score and Charlson score, with availability of MIMIC III dataset, we additionally obtained SOFA score and included in comparisons with other prediction models. We included in additional results of SOFA score in revised Table 2 and revised Figure 1 as well as the result section of the revised manuscript as suggested.

Table 2. Comparison of evaluation and calibration among the different models.

Model

Error Rate of Test Data Set

Accuracy

Precision

MCC

F1 score

AUROC in the Test Set

Brier score

Random forest model

21.4%

0.79

0.72

0.45

0.56

0.83 (0.79-0.87)

0.15

Decision tree

26.7%

0.73

0.59

0.30

0.44

0.71 (0.66-0.77)

0.19

Xgboost

25.0%

0.75

0.60

0.36

0.52

0.81 (0.76-0.85)

0.18

ANN

25.0%

0.75

0.67

0.33

0.42

0.79 (0.74-0.84)

0.19

Multivariable  logistic regression

22.9%

0.77

0.67

0.41

0.54

0.81 (0.79-0.83)

0.16

SOFA score

25.5%

0.74

0.67

0.30

0.39

0.74 (0.68-0.80)

0.17

SAPS II score

23.2%

0.77

0.71

0.39

0.49

0.77 (0.71-0.82)

0.17

Charlson score

28.4%

0.72

0.73

0.16

0.13

0.69 (0.63-0.74)

0.19

MCC: worst value –1 and best value +1. F1 score, accuracy, and precision: worst value 0 and best value 1. The Brier score is a combined measure of discrimination and calibration that ranges between 0 and 1, where the best score is 0 and the worst is 1.

Figure 1. Comparison of AUROC among the different ML models, logistic regression model, SAPS II score, Charlson score, SOFA score. (attached PDF in response to reviewer file)

Comment #5

The interpretability of the models should be further explored using LIME and Shapley.

Response:  The reviewer raises important point. We agree with the reviewer and thus we additionally applied SHAP and LIME to our RF model as the reviewer’s suggestions. The following has been added into the results or manuscript.

“To identify the features that influenced the prediction model the most, we depicted the SHAP summary plot of RF model (Figure 4) and the top 20 features of the prediction model. This plot depicts how high and low features’ values were in relation to SHAP values in the testing dataset. According to the prediction model, the higher the SHAP value of a feature, the higher probability of mortality occurs. Additionally, we applied LIME into RF model to illustrate the impact of key features at the individual level (Figure 5).”

Figure 4 (attached PDF in response to reviewer file). SHAP summary plot of the top 20 features of the RF model. The higher the SHAP value of a feature, the higher the probability of mortality. A dot is created for each feature attribution value for the model of individual patient. Each patient is allocated one dot on the line for each feature. Dots are colored according to the values of features for the respective patient and accumulate vertically to depict density. Red represents higher feature values, and blue represents lower feature values. Abbreviations: AKI, acute kidney injury; BUN, blood urea nitrogen; CRRT, continuous renal replacement therapy; HCO3, bicarbonate; ICU, intensive care unit; INR, international normalized ratio.

Figure 5 (attached PDF in response to reviewer file). Local interpretable model explainer (LIME) for 6 individual cases (case# 379 to 384) from the testing dataset. Label “1” means prediction of mortality and label “0” means prediction of no mortality (survival). Probability shows the probability of the observation belong to the label “1” or “0”. Five most influential variables that best explain the linear model in that observations local region are provided and whether the variable is causes an increase in the probability (supports/blue bar) or a decrease in the probability (contradicts/red bar). The x-axis shows how much each feature added or subtracted to the final probability value for the patient. Abbreviations: BUN, blood urea nitrogen; GCS, Glasgow Coma Scale; PTT, partial thromboplastin time; pO2, partial pressure of oxygen.

Thank you for your time and consideration.  We greatly appreciated the reviewer’s and editor’s time and comments to improve our manuscript. The manuscript has been improved considerably by the suggested revisions.

Reviewer 2 Report

The work presented is interesting. However, I have some suggestions to clarify some aspects of the work.

Specify the number of events per variable.
Put the TRIPOD checklist (https://www.equator-network.org/reporting-guidelines/tripod-statement/) in the supplementary files.

In addition to the AUC, present the calibration curves of all your models.
Have you tested the associations of the non-linear variables in the logistic regression model, e.g., with splines or a polynomial?

Provide more details on the different steps of optimizing the models and complete the information in the supplementary files, such as R scripts.

Suppose it is not possible to share your data to estimate individual risk. Is it possible to have the R computer file of the final model for the readers to use it in a standalone way on their PC or better an access to an R shiny?

Author Response

Response to Reviewer#2

Comment

The work presented is interesting. However, I have some suggestions to clarify some aspects of the work.

Response: We thank you for reviewing our manuscript and for your critical evaluation. We appreciate the reviewer’s valuable comments to improve our study. As suggested, we have additionally included TRIPOD checklist, calibration curves of all models, and detailed R scripts in supplementary data as suggestion. In addition, we applied SHAP and LIME approaches to our RF model for model’s explainability. As your important suggestion, we also successfully deployed our RF model to R Shiny as suggested: https://wisitc.shinyapps.io/RandomForestLacticAcid/

Comment #1

Specify the number of events per variable.

Response: The number of in-hospital mortality per variable was 25.4.

Comment #2

Put the TRIPOD checklist (https://www.equator-network.org/reporting-guidelines/tripod-statement/) in the supplementary files.

Response: We appreciate the reviewer’s important comment. We agree with the reviewer and we have included TRIPOD checklist in supplementary data as the reviewer’s suggested.

Comment #3

In addition to the AUC, present the calibration curves of all your models.

Response: We agree with the reviewer. We have additionally included all calibration curves of all models as suggested in supplementary Figure 4-10.

We have additionally included all calibration curves of all models as suggested in supplementary Figure 4-10.

Comment #4

Provide more details on the different steps of optimizing the models and complete the information in the supplementary files, such as R scripts.

Response: We agree with the reviewer. We have additionally included R scripts of hyperparameter optimizations in details in supplementary data as the reviewer’s suggestion.

Comment #5

Suppose it is not possible to share your data to estimate individual risk. Is it possible to have the R computer file of the final model for the readers to use it in a standalone way on their PC or better an access to an R shiny?

Response: We agree with the reviewer. Thus, as the reviewer’s suggestion, we successfully deployed our RF prediction model to R shiny https://wisitc.shinyapps.io/RandomForestLacticAcid/

The following text has been added as suggested.

Future validation external validation studies of ML prediction models are needed.

“In conclusion, a ML prediction model using the RF algorithm (available online as a shiny app at https://wisitc.shinyapps.io/RandomForestLacticAcid/) provided the highest predictive performance for hospital mortality among ICU patient with lactic acidosis. While future external validation studies are required, the findings of our study provide support towards the utilization of RF algorithms to improve risk stratification among critically patients with lactic acidosis.”  

Thank you for your time and consideration.  We greatly appreciated the reviewer’s and editor’s time and comments to improve our manuscript. The manuscript has been improved considerably by the suggested revisions.

Round 2

Reviewer 1 Report

My previous comments are well addressed.

Author Response

My previous comments are well addressed.

Thank you for your time and consideration.  We greatly appreciated the reviewer's and editor’s time and comments to improve our revised manuscript. The revised manuscript has been improved considerably by the suggested revisions.

Reviewer 2 Report

The authors have provided nice and appropriate answers. Thank you !

Author Response

The authors have provided nice and appropriate answers. Thank you !

Thank you for your time and consideration.  We greatly appreciated the reviewer’s time and comments to improve our revised manuscript. The revised manuscript has been improved considerably by the suggested revisions.
